# Tracking individual honeybees among wildflower clusters with computer vision-facilitated pollinator monitoring

**Malika Nisal Ratnayake**[1], **Adrian G. Dyer**[2,3], **Alan Dorin**[1]*

**1** Faculty of Information Technology, Monash University, Melbourne, Australia, **2** Department of Physiology, Monash University, Melbourne, Australia, **3** School of Media and Communication, RMIT University, Melbourne, Australia

* alan.dorin@monash.edu

**Data Availability Statement:** Software is available from GitHub: https://github.com/malikaratnayake/HyDaT_Tracker All training, experimental and video data available from the Monash Bridges data repository: https://bridges.monash.edu/articles/

## Abstract

Monitoring animals in their natural habitat is essential for advancement of animal behavioural studies, especially in pollination studies. Non-invasive techniques are preferred for these purposes as they reduce opportunities for research apparatus to interfere with behaviour. One potentially valuable approach is image-based tracking. However, the complexity of tracking unmarked wild animals using video is challenging in uncontrolled outdoor environments. Out-of-the-box algorithms currently present several problems in this context that can compromise accuracy, especially in cases of occlusion in a 3D environment. To address the issue, we present a novel hybrid detection and tracking algorithm to monitor unmarked insects outdoors. Our software can detect an insect, identify when a tracked insect becomes occluded from view and when it re-emerges, determine when an insect exits the camera field of view, and our software assembles a series of insect locations into a coherent trajectory. The insect detecting component of the software uses background subtraction and deep learning-based detection together to accurately and efficiently locate the insect among a cluster of wildflowers. We applied our method to track honeybees foraging outdoors using a new dataset that includes complex background detail, wind-blown foliage, and insects moving into and out of occlusion beneath leaves and among three-dimensional plant structures. We evaluated our software against human observations and previous techniques. It tracked honeybees at a rate of 86.6% on our dataset, 43% higher than the computationally more expensive, standalone deep learning model YOLOv2. We illustrate the value of our approach to quantify fine-scale foraging of honeybees. The ability to track unmarked insect pollinators in this way will help researchers better understand pollination ecology. The increased efficiency of our hybrid approach paves the way for the application of deep learning-based techniques to animal tracking in real-time using low-powered devices suitable for continuous monitoring.

dataset/Honeybee_video_tracking_data/12895433
DOI: https://doi.org/10.26180/5f4c8d5815940).

**Funding:** AD and AGD acknowledge the financial support of Australian Research Council Discovery Project Grant DP160100161 (www.arc.gov.au). The funders had no role in study design, data collection and analysis, decision to publish, or preparation of the manuscript.

**Competing interests:** Competing interests: Adrian G Dyer wishes to disclose that he is an editor for PLoS ONE.

## Introduction

Studying animal behaviour helps address key questions in ecology and evolution, however, collecting behavioural data is difficult [1]. While direct observation by ethologists is useful, this approach has low sampling resolution [2] and create bias due to attentional limitations [3], which makes it difficult to monitor fast moving animals such as insects [4]. Additionally, the accuracy of data may later be questioned since visual records of incidents are not preserved [5]. Video recordings potentially help overcome some methodological limitations by preserving observations. Unfortunately, manually extracting animal behaviour from video remains time consuming, and error prone due to the attentional limitations of human processing [3]. Recent advances in automated image-based tracking tackle these problems by extracting and identifying animal behaviours and trajectories without human intervention [5,6]. Whilst these techniques promise improved sampling of data, performance is still limited, in this case by environmental and animal behavioural complexity, and computational resources.

One area in which accurate, fine-scale behavioural data is particularly valuable is the study of insect pollination. Pollination is an integral requirement for horticulture and ecosystem management–insect pollinators impact 35% of global agricultural land [7], supporting over 87 food crops [8]. However, due to their small size and high speed whilst operating in cluttered 3D environments [4], insect pollinator monitoring and tracking is challenging. Since pollination is an ongoing requirement of crops and wildflowers alike, it would be ideal to establish field stations that can provide ongoing data on pollinator behaviours. To be practical, such a solution would need to be cheap to assemble and install. They would need to provide low cost, reliable and continuous monitoring of pollinator behaviour. These requirements exclude many current approaches to insect tracking, but the challenge is suitable for innovations involving imaging and AI.

Previous research has developed both invasive and non-invasive insect tracking methods. Invasive methods for example mark insects with electronic tags such as Passive Integrated Transponders (PIT) [9–12] or tags facilitating image-based tracking [13]. PIT-based tracking requires an electronic tag (e.g., harmonic radar, RFID) to be attached to an insect's body. Although, these methods can track insects over expansive areas and thus provide important larger scale information [14], the spatiotemporal resolution of collected data is lower than that of image-based tracking [5]. The latter approach is therefore better for data collection on fine insect movements likely to provide insight into cognition and decision making. Attaching tags to insects adds to their mass and may increase stress and alter behaviour [5,15,16], and, tagging individual insects is laborious, especially outdoors. For continuous season-long insect monitoring, attaching tags to populations of wild insects and managed honeybee hives containing potentially thousands of colony members is infeasible. Therefore, non-invasive methods such as unmarked image-based tracking can potentially make important contributions to our knowledge. Any improvements made to supporting technology can increase the scope and value of the approach.

Following unmarked insects is a difficult image-based tracking problem [17]. Previous tracking programs have been developed to research insect and small animal behaviour [17–22]. But their application is often confined to laboratories offering constant backgrounds and illumination needed for accurate tracking [17–21] or require human intervention [22]. Behavioural research on animals shows that environmental factors such as wind, temperature, humidity, sky exposure, may affect behaviour and interactions [23,24], and these are exactly the kinds of factors that field monitoring must explore. It is therefore essential to track insects outdoors in a biologically relevant scenario, rather than in a lab. In this study, we present novel

methods and algorithms to enable this. We illustrate the application of our methods by automatically tracking freely foraging honeybees.

Segmentation methods such as background subtraction and thresholding are widely used in image-based tracking to identify the position of animals in a video frame [17–19,25–28]. Background subtraction is efficient where background and illumination are constant, and significant background/object contrast exists [5]. This method has also been used to count and track honeybees [29–37] and bumblebees [1]. Most of this research to date has been conducted in laboratories, or in front of and within beehives with relatively constant backgrounds. This makes the application of pure background subtraction challenging.

Recently, there has been increased use of deep learning and neural networks for animal tracking [38]. Deep learning can detect and identify animals in a frame irrespective of the environment as it does not rely on foreground-background segmentation. The application of deep learning however has a high computational cost, and detection rate and accuracy depend on the quality and quantity of training data [39]. For rare species, or for species not previously tracked, a requirement for large training datasets increases the difficulty in implementing a tracking algorithm. Together, these factors currently limit the use of deep learning for generalised animal tracking, and for its application in remote devices for ecological research extracting movement and behavioural data from high-resolution data. Previous tracking approaches have used convolutional neural networks (CNNs) to estimate honeybee posture [25], distinguish between pollen-bearing and non-bearing honeybees [40], monitor interactions of honeybees in a hive [13] and monitor hive entry/exits [41]. However, taking steps towards the efficient and autonomous video tracking of unmarked insects in complex outdoor environments remains key to improving pollination and insect behavioural studies.

Insects forage amongst trees, leaves and flowers subject to changing illumination and movements caused by wind and animals (Fig 1). This increases tracking complexity [6] since the

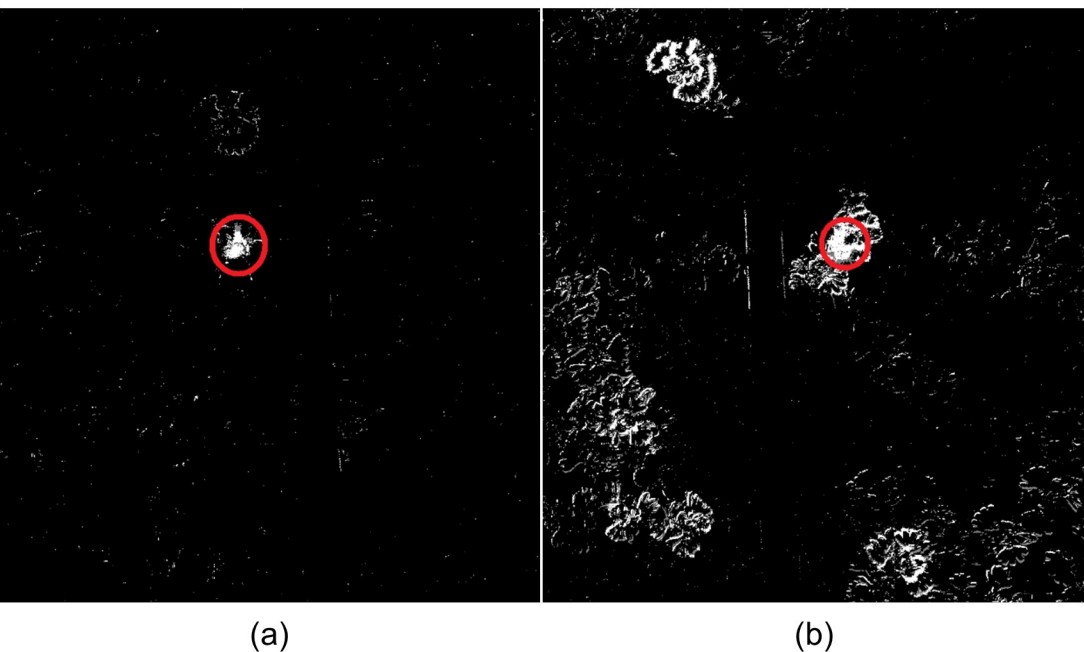

(a)                                                            (b)

**Fig 1. Foreground masks of an image showing a honeybee on a carpet of flowers obtained using background subtraction.**
The KNN background subtractor [48] was used to obtain foreground masks when the background is (a) constant; (b) wind-blown. Moving objects are shown in white pixels, the honeybee is circled.

changes detected in a frame of the video may relate to instances where one, the other, or both insect and non-insect elements (such as wind-blown leaves or flowers manipulated by an insect) in the environment move with respect to the camera. Ideally, it is desirable to detect an insect and identify its position in all of these scenarios to enable accurate census of pollinators, and what flowers they visit. Further complications arise as insects don't always fly, sometimes they crawl among and behind vegetation [42–44]. This can cause the insect to be occluded from view, or the insect may leave the camera's field of view completely resulting in frames where no position is recorded. To maintain the identity of the insect and terminate tracking if necessary, it is important for accurate recognition of insects as they move through a complex environment. Although previous research has tracked insects through occlusions in an open arena [45–47], identifying occlusions and view exits in unbounded, complex outdoor environments has not been previously reported.

In this paper, we present a novel Hybrid Detection and Tracking (HyDaT) algorithm to monitor foraging insects outdoors. Our purpose is to accurately map a sequence of interactions between a particular insect and its foraging environment. Hence, our implementation tracks one insect at a time from its entry to its exit from view, or from the start of a video sequence to the conclusion. In order to extract multiple plant-pollinator interaction sequences (actually, sequences of interactions between a unique pollinator and a set of flowers) we re-run the software on each insect detected in a region/clip in turn. To demonstrate our software in action, we train the detection model and tune parameters to track honeybees. We compare the efficiency and effectiveness of our algorithm against human ground observations and previously described methods, and apply our approach to track foraging on flower carpets in a new dataset (78 minutes of outdoor video). Finally, we discuss our results and suggest future improvements.

## Materials and methods

Our Hybrid Detection and Tracking (HyDaT) algorithm has four main components (Fig 2). A hybrid detection algorithm begins at the start of the video and moves through the footage until it first detects and identifies an as yet untracked insect. If this insect is not detected within a

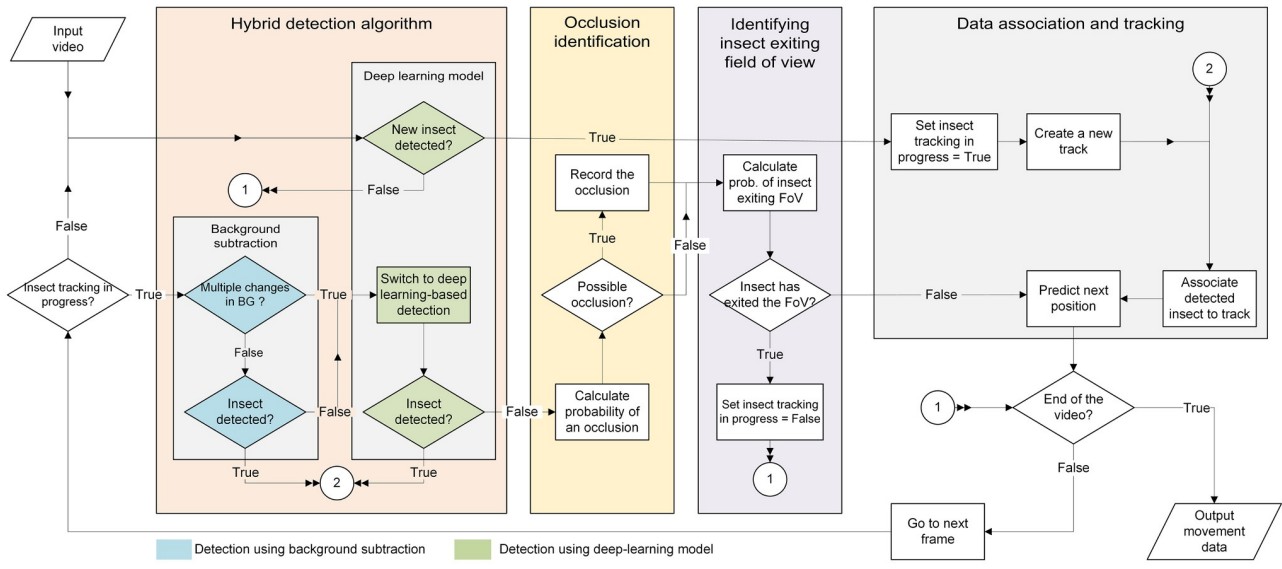

**Fig 2. Hybrid Detection and Tracking (HyDaT) algorithm overview and components.**

subset of subsequent frames, the algorithm uses novel methods to predict if it is occluded or has exited the view. Positional data collected from the algorithm is then linked to synthesise coherent insect trajectories. Finally, this information is analysed to obtain movement and behavioural data (e.g. heat-maps, speed or turn-angle distributions).

## The hybrid detection algorithm

We use a hybrid algorithm consisting of background subtraction and deep learning-based detection to locate an insect. As discussed in the introduction, background subtraction can detect movements in the foreground without prior training and works efficiently where the background is mainly stationary. In contrast, deep learning-based detection can detect and identify an insect irrespective of changes in the background, but it requires training with a dataset prior to use. We designed our hybrid detection algorithm to work with the strengths of each detection technique and intelligently switch between the two approaches depending on variations in the video's background. Prior to algorithm commencement, the deep-learning detection model must be trained on a dataset of the target insect species.

The algorithm begins using the trained deep learning model to initialise the detection process by locating the insect's first appearance in a video. This ensures identification of an insect with a low probability of false positives, even if the background is moving. After initial identification, the technique used for insect detection is determined by the number of regions of inter-frame change within a calculated radius $MDT_{DL}$ of the predicted position of the insect in the next frame (Data association and tracking, Eq 4). If there is a single region of significant change identified between frames, the background subtraction technique is used to locate the insect. If a small number of regions of change are detected within the predicted radius of the insect, then the region closest to the predicted position is recorded as the insect's position. (With our setup, three regions of movement within the calculated radius around the predicted position of the insect offered an acceptable compromise between algorithm speed and tracking accuracy. This trade-off can be user-adjusted). However, sometimes the region within the radius around the insect's predicted position is too full of movement to be sure which is the insect. In this case, background subtraction is unusable, or perhaps insufficiently inaccurate, so the hybrid algorithm switches to deep learning. In addition, whenever the background subtraction technique fails to detect movement likely to indicate the insect's position, deep leaning is used.

The hybrid detection algorithm consists of a modular architecture allowing state-of-the-art deep learning and background subtraction algorithm plug-ins to be incorporated as these tools advance. Details of deep-learning and background subtraction algorithms we use appear below.

**Deep learning-based detection.**   We use a convolutional neural network (CNN)-based YOLO (You Only Look Once) [49] object detection algorithm to detect insects in a video frame because it is well supported and convenient.

**Background subtraction-based detection.**   We use K-nearest neighbour (KNN)-based background/foreground segmentation [48] (OpenCV 3.4.1 [50]) to detect foreground changes in the video. The KNN background subtractor works by updating parameters of a Gaussian mixture model for better kernel density estimation [51]. The resulting binary image includes changes of the foreground assuming a constant background. A median filter and an erosion-based morphological filter are applied to the segmented image to remove noise. The resulting image contains changes in the foreground caused by insects and moving objects. Next, contours of the foreground detections (blobs) are extracted from the binary image and filtered based on their enclosing area to remove areas of movement less than a predetermined

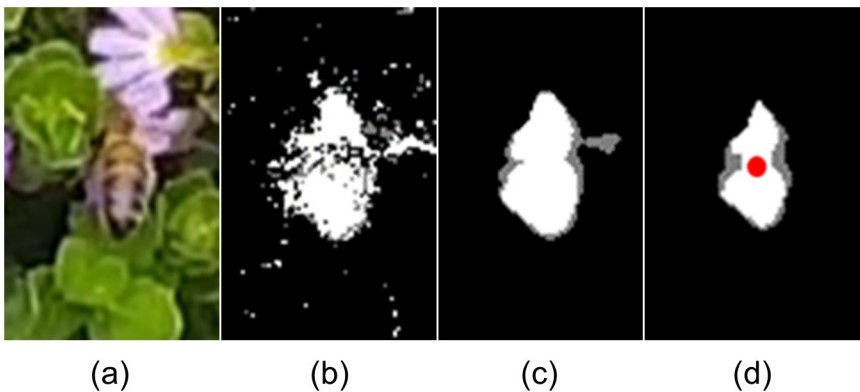

**Fig 3. Detecting an insect with background subtraction.** (a) Honeybee and flower shown at pixel resolution typical of that we employed for our study; (b) Binary image extracted using KNN background subtractor [48]; Resulting image with (c) median filter; (d) erosion-based morphological filter (centroid indicated).

minimum pixel count covered by the focal insect. The position of the insect is designated by the centroid of this filtered blob (Fig 3).

## Identifying occlusions

In the event that the focal insect is undetected, our algorithm analyses the variation in insect body area before its disappearance to identify a possible occlusion. Background subtraction is used to measure this change from the video. Variation of visible body area is modelled linearly using a least squares approach (Eq 1) to determine whether the insect is likely to have been occluded by moving under foliage.

$$m = \frac{n \sum Af - \sum A \sum f}{n \sum f^2 - \left(\sum f\right)^2} \tag{1}$$

Where $m$ m is the gradient of the linear polynomial fit, $n$ is the number of frames considered, $f$ is frame number, and $A$ is visible insect body area in frame $f$. When the insect crawls or flies under foliage, the variation of visible body area before disappearance shows a negative trend ($m < 0$). Our algorithm utilises this fact to identify whether the insect is occluded from view due to movement under foliage (Fig 4). If the insect disappears along a frame edge designating the camera's field of view, then the disappearance is assigned to a possible exit from the field of view, as discussed below. The algorithm for insect occlusion is not executed in this case.

## Identifying an insect exiting the field of view

To identify an insect's exit from view, we use Algorithm 1 to calculate an exit probability value $\beta$ when it has been undetected for a threshold of $\bar{\tau}$ consecutive frames. If $\beta$ is higher than a predefined threshold value $\bar{\beta}$, the algorithm pauses tracking the focal insect, and begins to search for new insects to track. If an insect is re-detected near the point of disappearance of the original focal insect before a new insect appears, the algorithm resumes tracking it, assuming this to be the original focal insect (see Discussion on Identity Swap management). Otherwise, the algorithm terminates and stores the previous track. Any new insect detections will be assigned to new tracks.

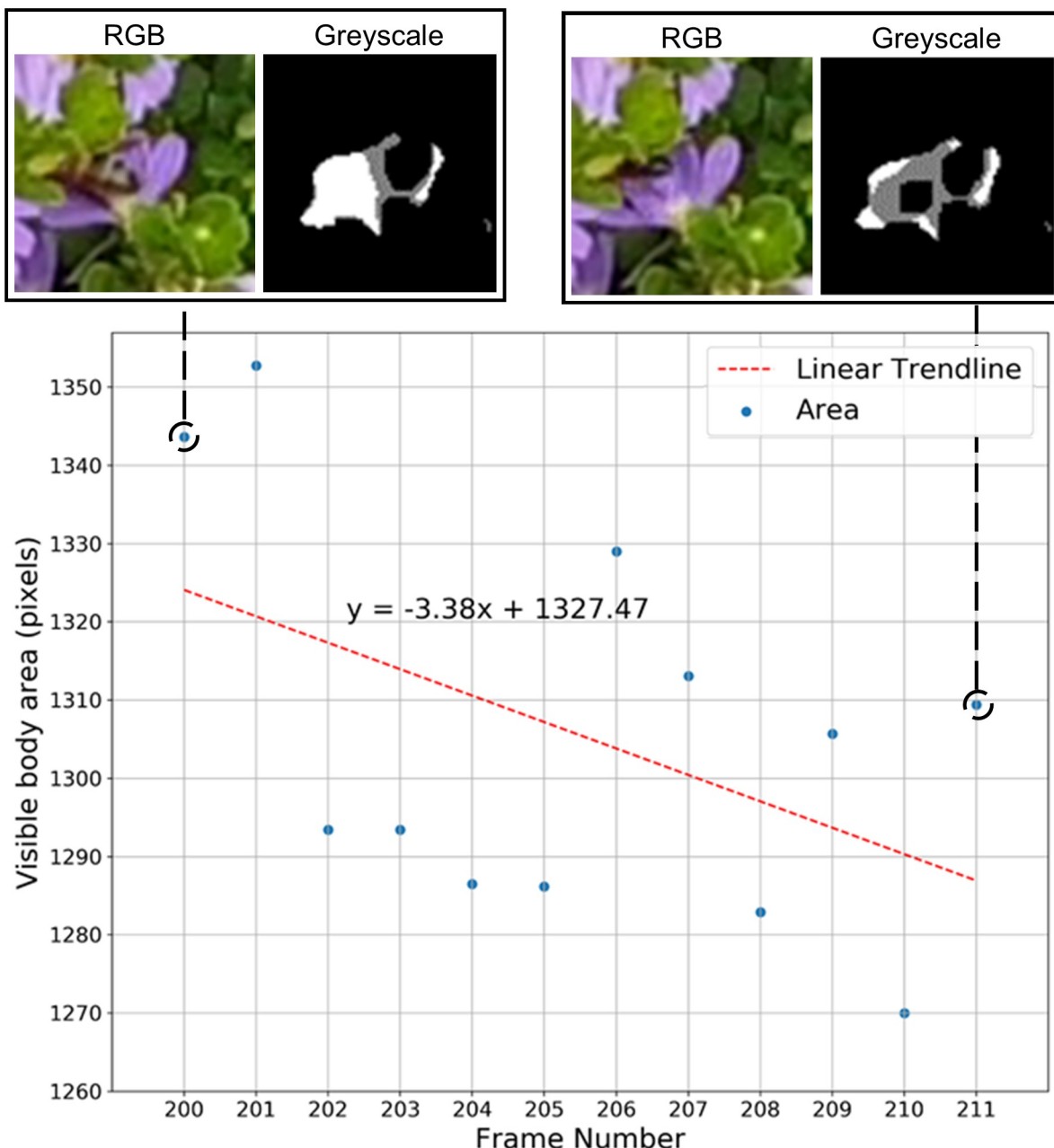

**Fig 4. An example of an insect occluded under foliage.** Scatterplot shows the variation of insect visible body area before occlusion, and the corresponding least squares polynomial fit. Pixel intensity in the greyscale image represents the amount of change detected in the foreground.

```
Algorithm 1: Calculating exit probability β
Input: Insect speeds, dₑ
Output: β
initialisation;
if τ = τ̄ then
i = 1.00;
else
i = last i;
```

```
end
dt = τ × ηi;
if dt > de then
while dt < de do
i- = 0.01:
dt = τ × ηi;
end
else
return;
end
β = (1 - i) × 100%
β Exit probability
de Shortest distance to frame boundary from insect's last detected
position
dt Predicted distance travelled by the insect during τ number of unde-
tected frames
τ Consecutive number of frames insect is not detected
τ̄ Threshold number of consecutive frames insect is not detected
i Quantile value
ηi ith quantile value of speed of the insect
```

## Data association and tracking

For applications discussed above, our algorithm tracks one insect at a time from its first appearance until its exit from view, before it is re-applied to track subsequent insects in footage. As a given frame may contain multiple insects simultaneously foraging in a region, a "predict and detect" approach is used to calculate the focal insect's track over successive frames. In a set of three successive frames, the predicted insect position in the third is calculated from the detected positions in the first two frames, assuming constant insect velocity over the three frames [35,52]. The predicted position $P_k$ of the insect in frame $k$ of the video is defined as:

$$P_k = [x_p k, y_p k]^T = A * [D_{k-1}, D_{k-2}]^T \qquad (2)$$

Where,

$$A = \begin{bmatrix} 2 & 0 & -1 & 0 \\ 0 & 2 & 0 & -1 \end{bmatrix}$$

In Eq (2) $x_p k$ and $y_p k$ refer to coordinates of the predicted position of the insect in the frame $k$ and $[D_{k-1}, D_{k-2}]$ are the detected positions of the insect in the two previous frames.

When an insect is first detected, the predicted position for the next frame is assumed to be the same as its current position (as there are no preceding frames). In the case of occlusions or frames in which no insect is detected, the predicted position is carried forward until the insect is re-detected.

In cases where multiple insects are detected within a single video frame using the hybrid algorithm, it is necessary to assign the predicted position of the focal insect to an individual detection within the frame. This is done using a process derived from the Hungarian method [53] which minimises the distance between assigned detections and predictions. To avoid recording false-positive detections, a detection is not associated with a prediction if the distance between the two surpasses a maximum threshold calculated using Eqs (3) and (4), based on distances travelled by the focal insect between consecutive frames within previously analysed data. Different detection thresholds are used for background subtraction ($MDT_{BS}$) and deep learning-based detection ($MDT_{DL}$) techniques, with $MDT_{BS} < MDT_{DL}$ since background

subtraction-based detections are more prone to false positives. Thresholds are defined as follows.

$$MDT_{BS} = \max\{d_{int}, \ d_{max}\} \tag{3}$$

$$MDT_{DL} = 2 \times \left( MDT_{BS} + \ \eta_{min\left\{\frac{\max\{0, \ (\tau - \ \bar{\tau})\}}{100}, \ 0.99\right\}} \right) \tag{4}$$

Where $d_{int}$ is the initial value for $MDT_{BS}$ set to the average body length (in pixels) of the target insect species, $d_{max}$ is the maximum recorded distance travelled by the focal insect between consecutive frames, $\eta_i$ is $i^{th}$ quantile of recorded speeds of the insect, $\tau$ is the number of consecutive frames during which the insect is not detected, and $\bar{\tau}$ is the predefined threshold number of consecutive frames during which the insect has gone undetected.

## Experiments and results

In this section, we evaluate the performance of our method (HyDaT) on honeybees (*Apis mellifera*). Honeybees are social insects that forage in wild, urban and agricultural environments. They are widespread, generalist pollinators of extremely high value to the global economy and food production [5], making honeybees particularly relevant organisms suited for testing our tracking.

We selected a patch of Scaevola (*Scaevola hookeri*) groundcover as the experimental site to evaluate our methods because of the species' tendency to grow in two dimensional carpets and to flower in high floral densities. Due to the undercover's structural density, honeybees both fly and crawl from flower to flower as they forage. Honeybees often crawl under the foliage to visit flowers that are obscured from above. These complexities in honeybee behaviour in Scaevola help us evaluate the robustness of our methods.

### Data collection for experiments

Videos required for experiments were recorded on the grounds of Monash University's Clayton campus, Melbourne, Australia (lat. 37.9115151˚ S, long. 145.1340897˚ E) in January 2019. All the videos were recorded between 10:00 am– 1:00 pm, ambient temperature $23˚C - 26˚C$, wind speeds $9 - 26 \ kmh^{-1}$. The study area contained ~446 flowers making a density of ~2340 $flowers/m^{-2}$. A Samsung Galaxy S8 phone camera (12 MP CMOS sensor, f/1.7, $1920 \times 1080$ pix, 60 fps) mounted on a tripod was set 600 $mm$ above the groundcover to record videos (Fig 5). A ruler placed in the recorded video frame was later used to convert pixel values to spatial scale (millimetres). Recorded videos covered an area of 600 $mm$ ×332 $mm$ with a density of 10.24 $pixels/mm^{-2}$. Average area covered by a honeybee was 1465 ± 531 $pixels$ (e.g. see Fig 3a).

### Software development

We developed the software using Jupyter Lab (Python 3.7.1), Computer Vision Library (OpenCV) 3.4.1 and Tensorflow 1.13.1. A Dell Precision 5530 workstation with Intel(R) Core i7-7820HQ (2.90 GHz) CPU, 32 GB Memory, 512 GB (SSD) storage and Microsoft Windows 10 Enterprise OS was used for processing. Data analysis was conducted using NumPy 1.16.2, Pandas 0.24.2 and Matplotlib 3.0.3. The code is available at github.com/malikaratnayake/HyDaT_Tracker.

A YOLOv2 object detection model [54] was used as the deep learning-based detection model in HyDaT. A Darkflow [55] implementation of YOLOv2 was trained using Tensorflow [56]. Images required for training the deep learning-based detection model were extracted from videos recorded in Scaevola groundcover using FrameShots [57]. Extracted images were

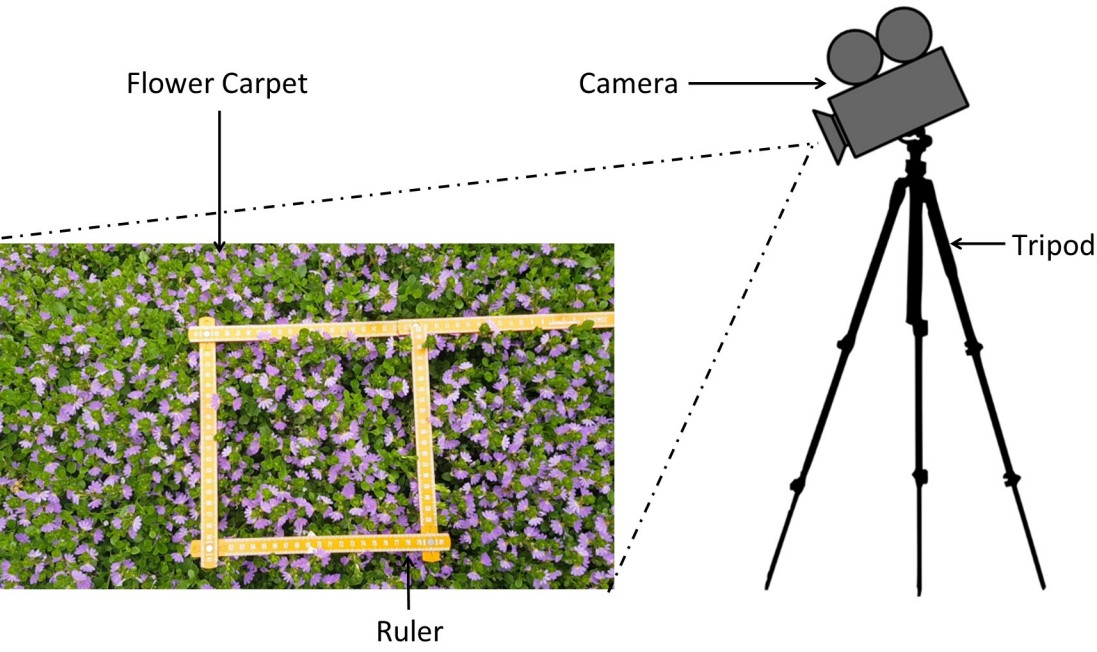

Flower Carpet

Camera

Tripod

Ruler

**Fig 5. Experimental setup for recording videos.**

then manually filtered to remove those without honeybees. The 2799 selected images containing honeybees were manually annotated with bounding boxes using LabelImg [58]. The annotated images and trained YOLOv2 model can be found online: https://doi.org/10.26180/5f4c8d5815940.

## Experiment 1: Detection rate and tracking time

We evaluated the detection rate and tracking time of HyDaT using a data set of seven video sequences of honeybees foraging in Scaveola. These videos were randomly selected from continuous footage of foraging honeybees. Each video was between 27 and 71 seconds long, totalling 6 minutes 11 seconds of footage in all. HyDaT was tuned to track the path of a honeybee from its first appearance in the video to its exit. All videos contained natural variation in background, lighting and bee movements. Fig 6 provides an explicit representation of each video sequence's changeability. One or more honeybee occlusions from the camera occurred in all of the videos.

*Detection rate* is our measure to evaluate the number of frames where the position of the insect is accurately recorded with respect to human observations. For the purpose of the experiment, frames where the honeybee is fully or partially hidden from the view were considered to be *occlusions*. If the algorithm recorded the position of the honeybee in an area that was in fact covered by the body of the bee, this was considered as a *successful detection*. The time taken by the algorithm to process the video was recorded as the *tracking time*.

We also compared the detection rate and tracking time of HyDaT to the stand-alone deep learning-based YOLOv2 [49] model after using the same training dataset for each. The aim of this was to evaluate the improvement in detection rate our methods can achieve compared to a deep-learning model under the same training regime and limitations. Parameters of our algorithm and the stand-alone YOLOv2 detection model were tuned separately to achieve maximum detection rates for each and allow it to operate at its best for comparison purposes

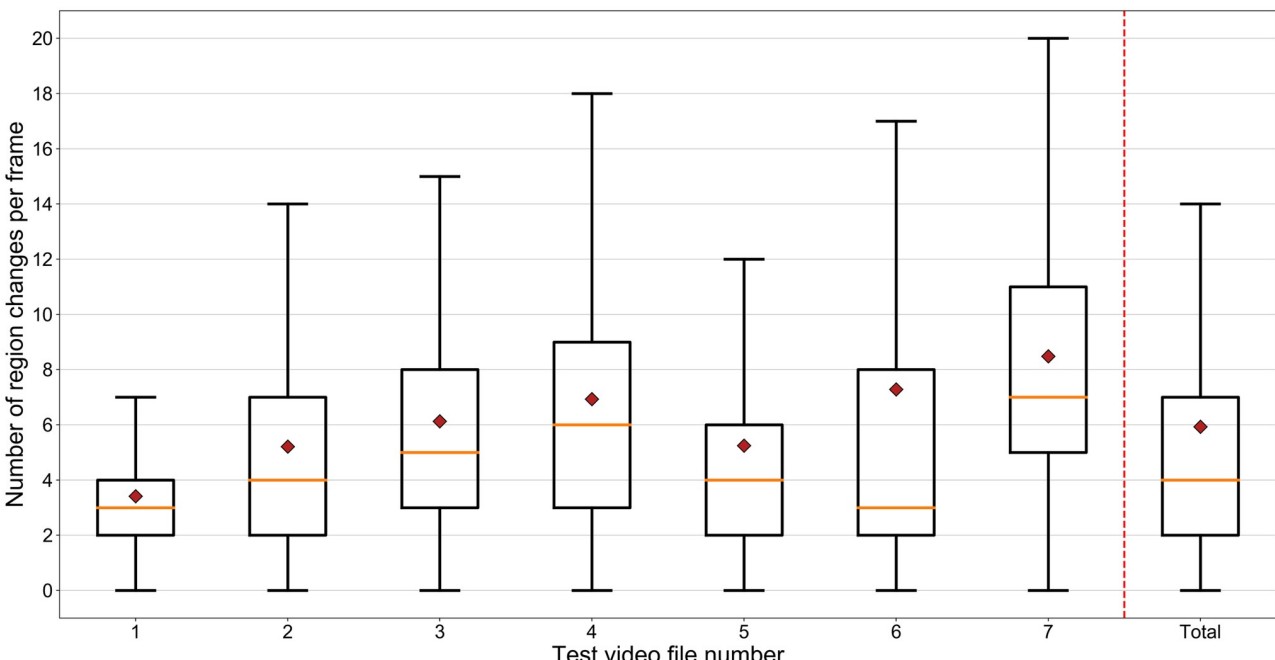

**Fig 6. Number of image region changes per frame in test videos.** Box plot showing the distribution of number of image regions with greater than one pixel change per frame in test videos. The filled red diamond indicates the mean number of region changes per frame.

(S1 Table). To benchmark our results further, we also processed the seven honeybee videos using Ctrax [18], current state-of-the-art insect tracking software.

Results are provided in Table 1. HyDaT detected the position of the honeybee and associated it to a trajectory in 86.6% of the frames in which it was visible to human observation. Compared to the stand-alone deep learning-based method YOLOv2 [49] model, HyDaT achieved higher detection rates for all seven test videos, a 43% relative increase in detection rate and a relative reduction in error of 66%. HyDaT processed the seven videos totalling 6 minutes 11 seconds (22260 frames at 60 fps) of footage in 3:39:16 hours, a reduction in

**Table 1. A quantitative comparison of HyDaTs' tracking performance against a stand-alone deep learning-based model (YOLOv2) [49] of honeybees foraging in Scaevola.**

| Video (Scaevola) | Number of frames | | Detection rate (%) | | Tracking time (hh:mm:ss) | | HyDaT's Detection method utilisation (%) | |
|---|---|---|---|---|---|---|---|---|
| | Video | Honeybee visible | HyDaT | YOLOv2 | HyDaT | YOLOv2 | Background Subtraction | Deep Learning (YOLOv2) |
| V1 | 3540 | 2670 | **97.7** | 76.4 | **00:29:29** | 01:18:26 | 94.9 | 5.1 |
| V2 | 2940 | 2148 | **51.5** | 36.9 | **00:45:45** | 01:03:55 | 95.8 | 4.2 |
| V3 | 3600 | 3016 | **91.9** | 63.1 | **00:33:51** | 01:19:20 | 85.2 | 14.8 |
| V4 | 2820 | 1612 | **72.6** | 50.1 | **00:38:16** | 00:53:05 | 98.7 | 1.3 |
| V5 | 3480 | 2802 | **89.2** | 26.9 | **00:30:26** | 01:05:40 | 94.5 | 5.5 |
| V6 | 4260 | 3882 | **97.1** | 84.0 | **00:28:58** | 01:22:11 | 87.6 | 12.4 |
| V7 | 1620 | 1414 | **89.1** | 77.3 | **00:12:32** | 00:33:17 | 88.2 | 11.8 |
| Overall | 22260 | 17544 | **86.6** | 60.7 | **03:39:16** | 07:35:54 | 91.0 | 9.0 |

Algorithm performance is assessed by detection rate (percentage of frames where the position of the honeybee accurately corresponds to human observations) and tracking time (the time taken to process a video). The "detection method utilisation" column shows the percentage of frames our algorithm used background subtraction versus deep learning methods to detect honeybee position. The best performing algorithm is indicated in bold.

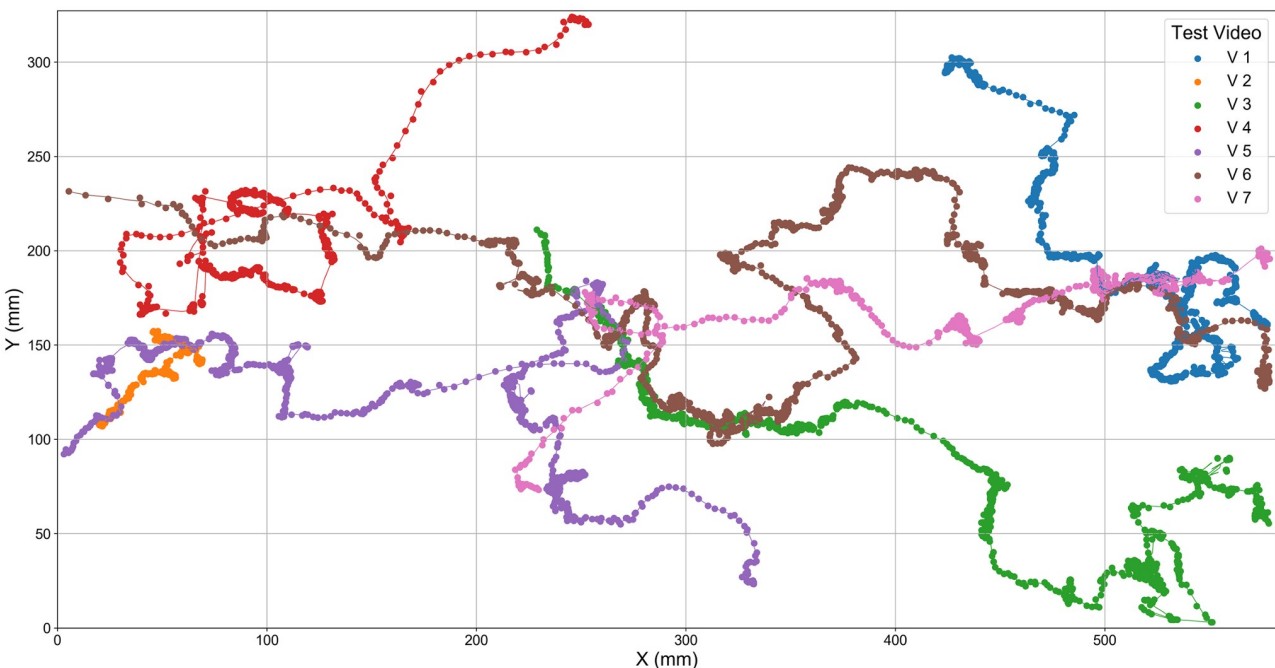

**Fig 7. Trajectories for a single honeybee in test videos.** Tracks were extracted using HyDaT from seven test video files.

tracking time of 52% over YOLOv2. This improvement in speed is possible because 91% of detections by HyDaT were made with background subtraction which requires much lower computational resources than purely deep learning based models. Ctrax, an existing animal tracking package we used for comparison, was completely unable to differentiate the movement of the honeybee from background movement. Its attempts to locate the honeybee were unusable and it would be meaningless to attempt to compare its results in these instances. In addition, when the honeybee was occluded for an extended period, Ctrax assumed it had left the field of view and terminated its track. Therefore, in these cases also it is meaningless to compare Ctrax's outputs with HyDaT. Tracks of honeybees extracted using HyDaT are shown in Fig 7.

### Experiment 2: Occlusion identification and exit frame estimation

The performance of the occlusion identification algorithm and the insect frame exit estimation were evaluated against human observations using a continuous video of duration 8 min. 15 sec. (29,700 frames) showing foraging honeybees in Scaevola. For this evaluation we only consider trajectories of bees visible for more than 120 frames (2 seconds at 60 fps). Threshold number of consecutive undetected frames, $\bar{\tau}$, was set to 15, and the threshold exit probability, $\bar{\beta}$, was 85%. The following guidelines were followed when conducting the experiment and determining the human ground observation values.

1. An insect was considered to be occluded from the view if it was partially or fully covered by a flower or a leaf and if it was not detected for over $\bar{\tau}$ frames.

2. An insect was considered to have exited the frame when it had completely left the camera view.

**Table 2. Occlusion detection algorithm performance and field of view (FoV) exit estimate for an 8:15 minute video of honeybees recorded in Scaevola.**

| Event | Actual no. of events | No. of events recorded (correct/incorrect) | No. of events missed | Detection rate (%) | Error in estimate (%) |
|---|---|---|---|---|---|
| Occluded | 35 | 24 (24/0) | 11 | 68.57 | 0.00 |
| Exited FoV | 16 | 19 (16/3) | 0 | 100.00 | 15.79 |
| Other | 3 | 11 (3/8) | 0 | 100.00 | 72.72 |

Detection rate = percentage of events correctly recorded compared to actual number of events; Error in estimate = percentage of incorrect recordings out of events recorded. "Other" in the Events column refers to instances where the insect was visible and the detection algorithm failed to locate it for over $\bar{\tau}$ (15) continuous frames.

Results are given in Table 2. The video evaluated for the study consisted of 54 instances where the honeybee was undetected by the software for over threshold value $\bar{\tau}$ (15) frames. The algorithm detected 68.57% of occlusions and all honeybee field of view (FoV) exits when compared to human analysis of the video.

### Example data analysis

To demonstrate the value of our approach for extracting meaningful data from bee tracks, we studied the behaviour of honeybees foraging in a Scaevola (*Scaevola hookeri*) as already discussed, and also in Lamb's-ear (*Stachys byzantine*) ground cover (Fig 8). We extracted spatiotemporal data of foraging insects and analysed their changes in position, speed and directionality. We tested our setup on both Scaevola and Lamb's-ear to assess the capability of our system to generalise, while simultaneously testing its ability to extend to tracking in three-dimensional ground cover, within the limits imposed by the use of a single camera.

We followed methods presented in Data collection for experiments section to collect study data. A dataset of 451 images was used to train the deep learning model of HyDaT on Lamb's-ear while the dataset used in experiments 1 and 2 was re-used for Scaevola (S1 Text). We extracted movement data from 38 minutes and 40 minutes of videos of honeybees foraging in Scaevola and Lamb's-ear respectively. Tracks longer than 2 seconds in duration were used for the analysis. Results of the study are shown in Fig 9.

Our algorithm was able to extract honeybee movement data in both two-dimensional (Scaevola) and three-dimensional (Lamb's-ear) ground covers. However, since our approach with a single camera is primarily suited to two-dimensional plant structures, the occlusion detection algorithm was unable to estimate the honeybee position in 36.5% of instances in the Lamb's-ear, compared to 8.8% of the instances in Scaevola (Fig 9c). We did not plot speed or

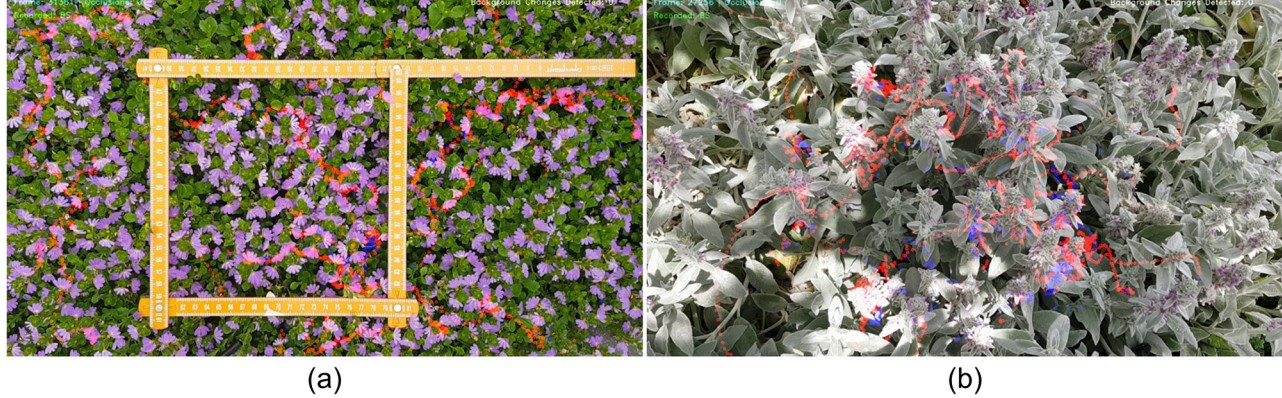

(a) (b)

**Fig 8. HyDaT algorithm tracking honeybee movement.** (a) Scaevola and (b) Lamb's-ear. Red indicates recorded positions.

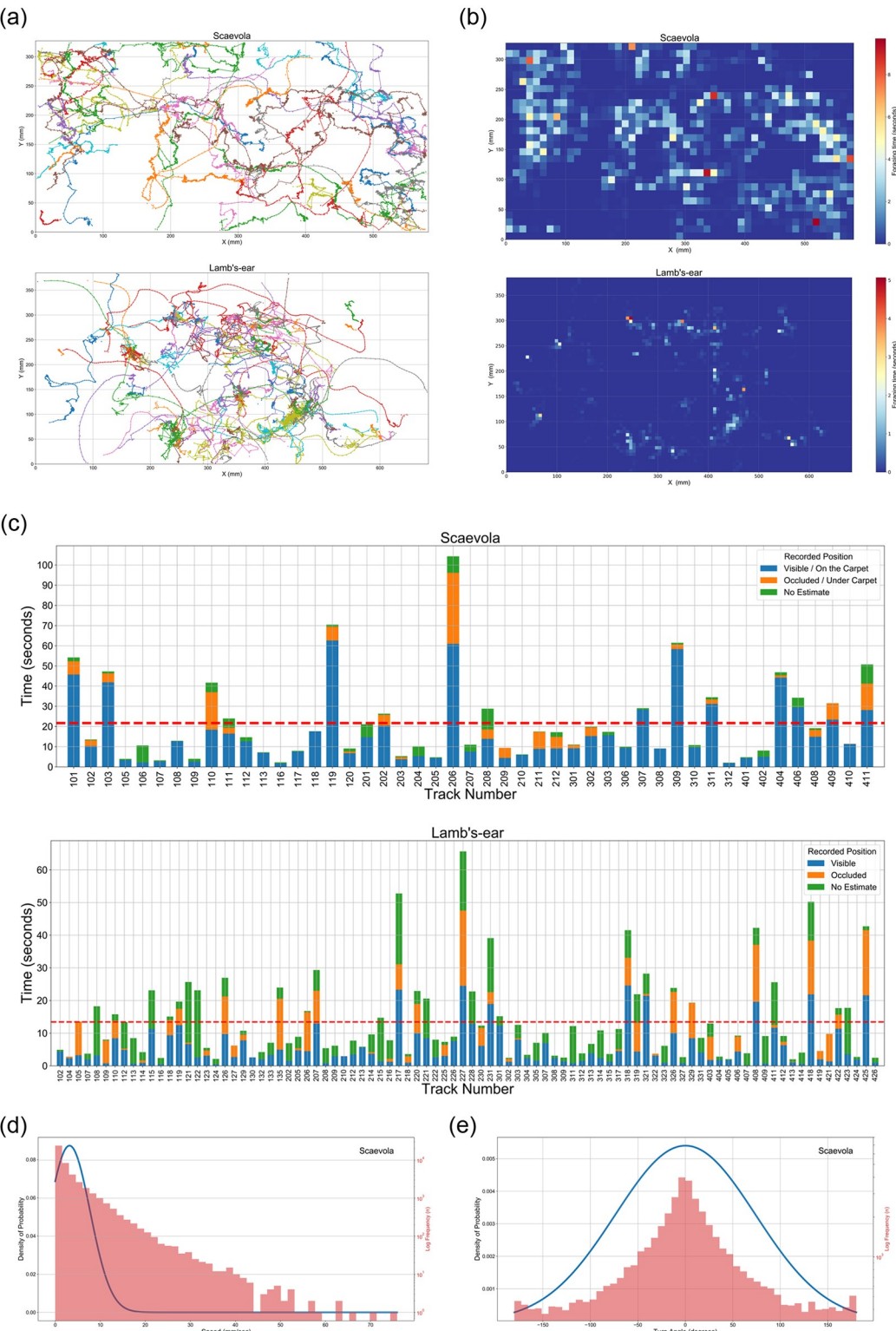

**Fig 9. Data analysis of honeybees foraging in Scaevola (N = 47) and Lamb's-ear (N = 90).** (a) Honeybee trajectories, (b) Location heat-maps, and (c) Visibility duration for Scaevola and Lamb's-ear. Honeybee (d) Speed distribution, (e) turn-angle distribution in Scaevola. In (b) the heat-map scale shows the aggregate of durations honeybees spent in a region. Bin size of the heat-map is the average area covered by a honeybee in pixels. In (c) recorded time is divided into durations the honeybee spends on the flower carpet (visible), under the carpet (occluded), and un-estimated, based on the output of the

occlusion identification algorithm. The red dashed line shows the mean foraging time of honeybees within the field of view of the camera.

turn-angle distributions for Lamb's-ear since a single camera setup cannot accurately measure these attributes for three-dimensional motion, a limitation we discuss below.

## Discussion

To address concerns about insect pollination in agriculture and ecosystem management, it is valuable to track individual insects as they forage outdoors. In many cases, such a capacity to work in real world scenarios necessarily requires handling data that includes movement of the background against which the insects are being observed, and movement of insects through long occlusions. We tackle this complexity using a novel approach that detects an insect in a complex dynamic scene, identifies when it is occluded from view, identifies when it exits the view, and associates its sequence of recorded positions with a trajectory. Our algorithm achieved higher detection rates in much less processing time than existing techniques.

Although we illustrated our method's generalisability in two differently structured ground covers, there remain several limitations associated with our method suited for further research. Our algorithm tracks one insect in sequence and must be restarted to track subsequent insects within a video. Future work could address this by considering models of multi-element attention [59], however this is unnecessary for the applications for which the software is currently being applied and was out of our scope. Regarding species other than honeybees; although we trained and tested our algorithm with honeybees as this is our research focus in the current study, tracking other species is feasible after retraining the YOLOv2 model and adjusting parameters for the area an insect occupies in the video frame and the $MDT_{BS}$, maximum detection threshold. Another potential subject for future study relates to identity swaps during occlusions, in which a single track is generated by two insects. This is likely to be a problem only in instances where insect densities are high and two insects cross paths, perhaps whilst occluded. Fingerprinting individual unmarked animals to avoid this is a complex image-based tracking problem [17,20,47] that, if solved, would enable such errors to be avoided. Previous research in the area has been conducted in controlled environments. Its application to the dynamic backgrounds necessary for our purpose of tracking insects in the wild will be challenging. Lastly, the accuracy of our single-camera method is diminished in three-dimensional plant structures such as the Lamb's Ear. Extending our method for multi-cameras would be worthwhile future work if insect behaviour within such plants was required for a particular study, although such solutions would increase cost base and complexity for surveying.

Our research's hybrid detection method combines existing background subtraction and deep learning-based detection techniques, to track honeybee foraging in complex environments, even with a limited training dataset. As applications of deep learning-based tracking is still relatively new to ecology, there is a scarcity of annotated datasets of insects. We also observe that the applicability of the datasets that are available currently to specific ecological problems will be dependent on the importance of the species documented and the environmental context in which the recordings were made. Therefore, in most instances ecologists will have to build and annotate new datasets from scratch to use deep learning-based tracking programs. Our methods will ease this burden on ecologists by enabling them to track insects with a relatively small training dataset.

Our algorithm is designed with a modular architecture, which enables any improvement to individual detection algorithms to be reflected in overall tracking performance. The current version of HyDaT was implemented with a KNN background subtractor and a YOLOv2

detection model. However, use of different combinations of detection models for background subtraction and for deep learning models may further improve detection rates and tracking speeds. This allows ecologists to quickly adopt advancements in deep learning or computer vision research for improved tracking.

Mapping interactions between insect pollinators and their foraging environments improves our understanding of their behaviour. Previous research has studied the movement patterns of insect pollinators such as honeybees [44,60–63] and bumblebees [42–44,62,64,65] to document their flight directionality, flight distance, time on a flower, nature of movement etc. Most of this research relied on manual observations conducted inside laboratories or on artificial rigs. However, environmental factors such as wind, temperature and other conditions may play a role in driving insect behaviour outdoors [23,24]. Our tracking method facilitates researchers to study insect pollinators in their natural habitat and enables collection of accurate, reliable data. This capacity may be expanded across a network of monitoring sites to assist in the automatic measurement of behavioural traits such as flower constancy of bees in complex environments [66]. In addition, our algorithm can record when insects crawl under flowers, a frequent occurrence that previous algorithms have not considered.

Commercial crops such as strawberry, carrot and cauliflower flower in a somewhat flat carpet of inflorescences when compared against other insect-pollinated crops such as raspberry and tomato. Our algorithm is particularly suited to record and analyse the trajectories of insect pollinators on such two-dimensional structures and can therefore be used to monitor agricultural insect pollination in these circumstances. This enables growers and beekeepers to estimate pollination levels and take proactive steps that maximise pollination for better crop yield [67]. We hope that ultimately these findings will be helpful in pollinator conservation and designing pollinator-friendly agricultural setups[67] for increased food production.

While our main contribution is tracking insect pollinators in complex environments, our results are an important step towards real-time tracking and implementing deep learning-based object detection models in low powered devices such as the Raspberry Pi (www. raspberrypi.org) which are suited to ongoing field monitoring of insect populations and behaviours. Through experiments, we have shown that combining computationally inexpensive detection methods like background subtraction with deep learning can increase the rate of detection and reduce computational costs. Hence, our hybrid approach may be suited to applications where low-powered devices should be used.

## Supporting information

**S1 Table. Parameter settings used in experiments.**
(DOCX)

**S1 Text. Data collection details for experimental data analysis.**
(DOCX)

## Acknowledgments

Authors would like to thank Prof. Dinh Phung for the valuable suggestions and helpful comments given in designing the methods and experiments and Dr. Mani Shrestha for assisting in species identification.

## Author Contributions

**Conceptualization:** Malika Nisal Ratnayake, Adrian G. Dyer, Alan Dorin.

**Data curation:** Malika Nisal Ratnayake.

**Formal analysis:** Malika Nisal Ratnayake.

**Funding acquisition:** Adrian G. Dyer, Alan Dorin.

**Investigation:** Malika Nisal Ratnayake.

**Methodology:** Malika Nisal Ratnayake.

**Project administration:** Alan Dorin.

**Resources:** Alan Dorin.

**Software:** Malika Nisal Ratnayake.

**Supervision:** Adrian G. Dyer, Alan Dorin.

**Validation:** Malika Nisal Ratnayake.

**Visualization:** Malika Nisal Ratnayake.

**Writing – original draft:** Malika Nisal Ratnayake.

**Writing – review & editing:** Malika Nisal Ratnayake, Adrian G. Dyer, Alan Dorin.

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
