## [Decision Letter · Decision Letter 0]

19 Jan 2021

Tracking individual honeybees among wildflower clusters with computer vision-facilitated pollinator monitoring

PONE-D-20-27840

Dear Dr. Dorin,

We’re pleased to inform you that your manuscript has been judged scientifically suitable for publication and will be formally accepted for publication once it meets all outstanding technical requirements.

Kind regards,

Nicolas Chaline

Academic Editor

PLOS ONE

Additional Editor Comments (optional):

Reviewers' comments:

Reviewer's Responses to Questions

**Comments to the Author**

1. Is the manuscript technically sound, and do the data support the conclusions?

Reviewer #1: Yes

2. Has the statistical analysis been performed appropriately and rigorously? 

Reviewer #1: N/A

3. Have the authors made all data underlying the findings in their manuscript fully available?

Reviewer #1: Yes

4. Is the manuscript presented in an intelligible fashion and written in standard English?

Reviewer #1: Yes

5. Review Comments to the Author

Reviewer #1: The authors developped an hybrid algorithm (background susbtraction and deep learning detection) for computer vision. They show how this can be used to track individual honey bees in natural flower patches. I have little to say. This is a nicely written method paper that will certainly be useful for future ecological studies on bee behaviour. As far as I understand the approach is valid and deserves publication. The main limitations are carefully explained.

6. PLOS authors have the option to publish the peer review history of their article (what does this mean?). If published, this will include your full peer review and any attached files.

Reviewer #1: **Yes: **Mathieu Lihoreau

---

## [Editor Report · Acceptance letter]

25 Jan 2021

PONE-D-20-27840 

Tracking individual honeybees among wildflower clusters with computer vision-facilitated pollinator monitoring 

Dear Dr. Dorin:

I'm pleased to inform you that your manuscript has been deemed suitable for publication in PLOS ONE. Congratulations! Your manuscript is now with our production department. 

Kind regards, 

on behalf of

Professor Nicolas Chaline 

Academic Editor

PLOS ONE